# Short Signature *rpoB* Gene Sequence to Differentiate Species in *Mycobacterium abscessus* Group

Jian R. Bao,[a] Kileen L. Shier,[a] Ronald N. Master,[a] Robert S. Jones,[a] Richard B. Clark[a]

aQuest Diagnostics Nichols Institute, Chantilly, Virginia, USA

**ABSTRACT** *Mycobacterium abscessus* group (MAG) are rapidly growing acid-fast bacteria that consist of three closely related species: *M. abscessus* (*Ma*), *M. bolletii* (*Mb*), and *M. massiliense* (*Mm*). Differentiation of these species can be difficult but is increasingly requested owing to recent infectious outbreaks and their differential drug resistance. We developed a novel and rapid pyrosequencing method using short signature sequences (35 to 45 bp) at a hypervariable site in the *rpoB* gene to differentiate the three MAG species, along with *M. chelonae* (*Mc*), and *M. immunogenum* (*Mi*). This method was evaluated using 111 *M. chelonae-abscessus* complex (MCAC) isolates, including six reference strains. All isolates were successfully differentiated to the species level (69 *Ma*, four *Mb*, six *Mm*, 23 *Mc*, and nine *Mi*). The species identifications by this method had 100% agreement with Sanger sequencing as well as an *in-silico rpoB* typing method. This short signature sequencing (SSS) method is rapid (6 to 7 h), accurately differentiates MAG species, and is useful for informing antimicrobial therapy decision.

**IMPORTANCE** *Mycobacterium abscessus* group (MAG) are rapidly growing acid-fast bacteria that include three species: *M. abscessus, M. massiliense*, and *M. bolletii*. These species are among the leading causes of nontuberculosis mycobacteria infections in humans but difficult to differentiate using commonly used methods. The differences of drug resistance among the species shape the treatment regimens and make it significant for them to be differentiated accurately and quickly. We developed and evaluated a novel short signature sequencing (SSS) method utilizing a gene called *rpoB* to differentiate the three MAG species, as well as other two species (*M. chelonae* and *M. immunogenum*). The identification results had 100% agreement with both the reference method of Sanger sequencing and *rpoB* typing method via a computer-simulated analysis. This SSS method was accurate and quick (6 to 7 h) for species differentiation, which will benefit patient care. The technology used for this method is affordable and easy to operate.

**KEYWORDS** *Mycobacterium abscessus* group (MAG), short signature sequencing (SSS), species differentiation, *rpoB* gene, *M. chelonae-abscessus* complex

**M**ycobacterium (*Mycobacteroides*) *abscessus* group (MAG) is one of the leading causes of nontuberculosis mycobacteria (NTM) infections in humans, which include soft tissue, respiratory, and central nervous system infections (1–4). The organisms in this group are rapidly growing acid-fast bacteria (AFB) and composed of three species: *M. abscessus*, *M. massiliense*, and *M. bolletii*, which are virtually indistinguishable phenotypically. In the United States, the MAG species comprise 2.6% to 13% of all mycobacterial infections (4). They were the etiological agents for major NTM outbreaks in Brazil from 2004 to 2008, predominantly caused by *M. massiliense* and *M. bolletii* (1, 5–7), in the United States and other parts of the world (8, 9). The characteristic drug resistance among the MAG species is a challenge for therapy. *M. abscessus* and *M. bolletii* are resistant to macrolides owing to the possession of a functional erythromycin ribosome methyltransferase gene *erm(41)*, while *M. massiliense* possesses a truncated gene

Address correspondence to Jian R. Bao, jian.r.bao@questdiagnostics.com.

The authors declare no conflict of interest.

that is nonfunctional (10, 11). The resistance differences shape the treatment regimens among the species-specific infections, and recent outbreaks signify the need for the species differentiation in patient care (3, 6, 9, 10).

In clinical laboratories, MAG species are commonly reported as *M. chelonae-abscessus* complex (MCAC), which includes the MAG, *M. chelonae* group, and other related species that cannot be differentiated phenotypically or by 16S rRNA gene sequencing (12). MCAC has 11 species or subspecies, but the three MAG species (*M. abscessus*, *M. massiliense*, and *M. bolletii*), plus *M. chelonae*, and *M. immunogenum*, are the five most-well-characterized opportunistic human pathogens. The other six MCAC taxa are most often recovered from water-related environments and their clinical relevance to humans remains largely unclear (3). In this study, we focused on development of a simple molecular method to differentiate the five clinically relevant species.

Many methods have been evaluated for differentiating the MAG species or other MCAC species. Conventionally, isolates were tested for their ability to grow on 5% NaCl solid medium in 2 to 4 weeks to differentiate the salt-tolerant MAG from *M. chelonae* group, but the salt tolerance test is not reliable and does not differentiate the individual species (13, 14). Today, matrix-assisted laser-desorption/ionization-time of flight (MALDI-TOF) mass spectrometry can differentiate the *M. abscessus* group from the *M. chelonae* group, but not very successfully to the species level (15–17). Whole-genome or multilocus (such as *hsp*65, *rpoB*, *secA*) sequencing are now the standard to define microbial taxa but are rarely used in most clinical laboratories because of the high cost and technical expertise required. Sequencing of the 16S rRNA gene is widely used to identify *Mycobacterium* species, but the gene lacks differential base-pairs for MCAC species (12, 18). Most recently, GenoType NTM-DR line probe assay (Hain Lifescience, Nehren, Germany) has been used in some clinical laboratories to differentiate NTM species or subspecies, including the MAG species (19, 20). Signature sequence DNA hybridization to genomes and a two-tube-based real-time PCR method showed promise in identifying the three MAG species (21, 22). The method we developed in this study adds another quick and useful tool for the MAG species differentiation.

In this study, we used the RNA polymerase beta subunit gene (*rpoB*) to develop a rapid and accurate method to identify these organisms. The gene is a well-known drug-resistance analysis target in mycobacteria. It has more differential power in species identification and has been used for novel taxa classifications (1, 14, 23, 24). Pyrosequencing technology has been used in clinical laboratories to identify and detect bacterial organisms and is quicker, simpler, and cost-effective when compared with other available sequencing methods (18, 25–28). Here, we developed and evaluated a short signature sequencing (SSS) method using the *rpoB* gene and pyrosequencing technology to differentiate MAG species in the clinical laboratory.

## RESULTS

**The short signature sequence in the *rpoB* gene hypervariable site.** The developed assay produced strong PCR product bands having correct DNA fragment sizes (about 360 bp) on electrophoresis gel. The PCR product was relatively long for this technology as closer optimal primer binding sites were not found during our *in-silico* analysis, but no negative impact was observed during the assay operation. The signature pyrosequencing products had strong sequencing signals on the assay system for all the tested isolates and were typically 35 to 45 bp long depending on the species. A typical assay run from DNA extraction to final species determination took 6 to 7 h.

Though the length of short signature sequences varies slightly with different species, we took a typical 35-bp sequence as a representative for each of the five targeted species and examined the numbers of differential nucleotide base pairs among the species. There were two to 11 discriminating base pairs available for differentiation of the five species (Fig. 1). Within the three MAG species, *M. abscessus* was separated from *M. bolletii* by 10 bp differences (71% homology), and from *M. massiliense* by 8 bp differences. *M. bolletii* was separated from *M. massiliense* by 3 bp differences. The three

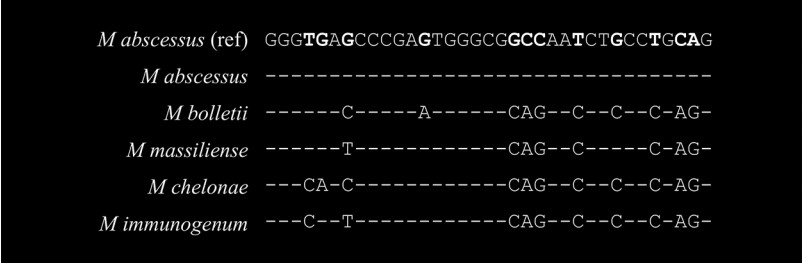

**FIG 1** Typical pyrogram-generated signature *rpoB* sequences of the five species cited from their alignments to the reference sequence (ref) from *M. abscessus* strain CIP 104536 (T); –, indicates the same nucleotide as that in the reference sequence. Bolded font in the reference sequence indicates variable nucleotides.

MAG species had two to 11 differential bps available to separate either *M. chelonae* or *M. immunogenum*, while the species *M. abscessus* itself had 10 to 11 differential base pairs for separation from the two non-MAG species (*M. chelonae* or *M. immunogenum*). *M. chelonae* was separated from *M. immunogenum* by two transitional bp changes in the typical signature sequences, while most of the base pair changes among the species were caused by nucleotide transversion.

**MAG species identification evaluation.** This SSS method identified all 111 MCAC isolates, including all six reference strains, to the five targeted species (Tables 1 and 2). Seventy-nine (71%) of isolates were MAG species, and *M. abscessus* was the predominant species in the group (69 of the 79 isolates, 87%). *M. massiliense* and *M. bolletii* were six (8%) and four (5%) of isolates, respectively. Twenty-three (21%) were *M. chelonae* group and nine (8%) were *M. immunogenum*. As the clinical isolates were randomly selected from patient cultures, the isolate number ratios of each species at some degree reflected the species distributions for the MCAC and MAG in the eastern U.S. regions. *M. abscessus* was the dominant species in both the complex (62%) and the MAG. Both this *rpoB* short signature sequencing method and our existing 16S rRNA gene pyrosequencing method were able to identify the species *M. immunogenum*.

The reference strain ATCC 700868 was received as *M. abscessus* (ATCC, Manassas, VA). The SSS method identified this strain as *M. bolletii*. The identification as *M. bolletii* was confirmed by our Sanger sequencing method and was further validated to be correct by a multigene-based line-probe assay (Hain Life Science, Germany) performed at Advanced Diagnostic Laboratories, National Jewish Health (Denver, CO).

**Validation by Sanger sequencing and other methods.** The Sanger sequencing produced approximately 690 bp sequences from the PCR products after the trimming of the primer nucleotides at both ends of the amplicons. The Sanger sequencing method produced the same identification results (100% agreement) as the SSS method on the tested 40-isolate subset, including all six reference strains (Table 3). The sequence homologies to their reference sequences from respective type strains were matched from 99.5% to 100%. The distance scores to their next closest species among the species identified from the Sanger sequencing method was 0.5% at minimum with range of 0.5% to 2.8%, which met or exceeded the correct species identification

**TABLE 1** Total MCAC isolates tested for species identification using the short signature *rpoB* gene sequencing method

| Complex (no.) | Group (no.) | Species identified | No. (% to total) |
|---|---|---|---|
| *M. chelonae-abscessus* Complex (MCAC) (111) | *M. abscessus* group (MAG) (79 or 71%) | *M. abscessus* | 69 (62) |
| | | *M. massiliense* | 6 (5) |
| | | *M. bolletii* | 4 (4) |
| | Others (32) | *M. chelonae* | 23 (21) |
| | | *M immunogenum* | 9 (8) |

**TABLE 2** Reference strain identification using the short signature *rpoB* gene sequencing (SSS) method

| Strain (origin) | Organism species | MALDI-TOF for Group ID[c] | LJ salt[a] | SSS method |
|---|---|---|---|---|
| ATCC 35752 | *M. chelonae* | *M. chelonae* | − | *M. chelonae* |
| ATCC 19536 | *M. chelonae* | *M. chelonae* | + | *M. chelonae* |
| ATCC 35749 | *M. chelonae* | *M. chelonae* | NT | *M. chelonae* |
| 26811 (Survey) | *M. chelonae* | *M. chelonae* | NT | *M. chelonae* |
| 27369 (Survey) | *M. bolletii* | *M. abscessus* | NT | *M. bolletii* |
| ATCC 700868 | *M. abscessus* | NT[b] | + | *M. bolletii*[d] |

[a]+, positive; −, negative or no growth.
[b]NT, not tested.
[c]ID, identification.
[d]The identification of *M. bolletii* was confirmed by using a line-probe analysis (Advanced Diagnostic Laboratories, National Jewish Health, Denver, CO).

requirement (29). The Sanger sequencing results validated that the species identifications by this SSS method were correct.

MALDI-TOF identified the isolates to either MAG or *M. chelonae* group (MCG) but did not further differentiate to their individual species (Table 3). When the tube extraction procedure was used, MALDI-TOF did not demonstrate either labor or turnaround time savings compared with this SSS method. The salt tolerance method is a group-based separation method and produced 12.5% incorrect results for the group differentiations and 4% of indeterminate results among the 50 isolates tested.

***In-silico* analysis.** Among the 95 strains randomly selected from the curated organism pool for *in-silico* analysis (22), the SSS method produced 100% agreement with the average nucleotide identity (ANI) or *rpoB* typing methods for identifications of both the six respective type strains and the additional 89 well-characterized strains of the three MAG species (Table 4; Table S1 and S3). In addition, all the *in-silico* short signature sequences had 100% homologies to those reference sequences in MAG species. The analysis results from the type strains of *M. chelonae*, *M. immunogenum*, and *M. phocaicum* also had 100% agreement of both their identifications and sequence matches (Tables S1 and S3). The *in-silico* results, plus the Sanger sequencing results as described above, demonstrated that the SSS method was accurate to identify the MAG species and our in-house database was valid for the SSS method.

Compared with ANI method from the 1505 MAG strain pool (22), the *ropB* typing method had 98.5% to 100% identification agreements depending on the MAG species, while the *erm41* typing method had 97.0% to 100% agreements (Table 4B; Table S2). The discrepancies were from either *M. abscessus* or *M. massiliense*, not from *M. bolletii*, though the *erm41* typing method grouped *M. abscessus* and *M. bolletii* as *M. abscessus/bolletii*.

**TABLE 3** Organism identifications from different methods in comparison with the SSS method[a]

| SSS method ID[b] (#111) | Sanger sequencing method (#40) | | MALDI-TOF[c] method (#42) | | Growth on LJ[d] + 6% Salt (#50) | | | | 16S rRNA gene method (#111) |
|---|---|---|---|---|---|---|---|---|---|
| | Organism ID | Agree (%) | Organism ID | Overall agree (%) | No isolates tested | MAG (growth) | MCG (no growth) | In-determinate (no ID) | Organism ID |
| *M. abscessus* | *M. abscessus* | 100 | MAG[f] | 76.2 | 30 | 27 | 1 | 2 | MCAC[e] |
| *M. chelonae* | *M. chelonae* | 100 | MCG[g] | | 8 | 5 | 3 | 0 | MCAC |
| *M. massiliense* | *M. massiliense* | 100 | MAG | | 5 | 3 | 0 | 2 | MCAC |
| *M. bolletii* | *M. bolletii* | 100 | MAG | | 3 | 1 | 0 | 2 | MCAC |
| *M. immunogenum* | *M. immunogenum* | 100 | MCG | | 4 | 0 | 1 | 3 | *M. immunogenum* |

[a]SSS, short signature sequencing.
[b]ID, identification.
[c]MALDI-TOF, matrix-assisted laser desorption/ionization–time-of-flight mass spectrometry.
[d]LJ, Lowenstein-Jensen slants.
[e]MCAC, *M. chelonae-abscessus* complex.
[f]MAG, *M. abscessus* group.
[g]MCG, *M. chelonae* group.

**TABLE 4** *In-silico* analysis for this SSS method and its comparison with other methods

| Organism | Tested no. | No. of discrepancy | Identification agreement (%) | Sequence homology (%) |
|---|---|---|---|---|
| A. Analysis from this study | | | | |
| SSS identification compared with their type strains | | | | |
| *M. abscessus* | 69 | 0 | 100 | 100 |
| *M. bolletii* | 4 | 0 | 100 | 100 |
| *M. massiliense* | 6 | 0 | 100 | 100 |
| SSS method compared with ANI or *rpoB* typing methods | | | | |
| *M. abscessus* | 37 | 0 | 100 | 100 |
| *M. bolletii* | 16 | 0 | 100 | 100 |
| *M. massiliense* | 42 | 0 | 100 | 100 |
| B. Analysis from reported data for method comparisons[a] | | | | |
| ANI method compared with *rpoB* typing method | | | | |
| *M. abscessus* | 941 | 8 | 99.2 | NA[c] |
| *M. bolletii* | 110 | 0 | 100 | NA |
| *M. massiliense* | 454 | 7 | 98.5 | NA |
| ANI method compared with *erm41* typing method[b] | | | | |
| *M. abscessus* | 941 | 28 | 97.0 | NA |
| *M. bolletii* | 110 | 0 | 100 | NA |
| *M. massiliense* | 454 | 11 | 97.6 | NA |
| *rpoB* typing compared with both ANI method and *erm41* typing[b] | | | | |
| *M. abscessus* | 939 | 6 | 99.4 | NA |
| *M. bolletii* | 110 | 0 | 100 | NA |
| *M. massiliense* | 456 | 2 | 99.6 | NA |

[a]Reported data were based on the summary by Minias et al. (22).
[b]*M. abscessus* and *M. bolletii* were grouped as *M. abscessus/bolletii* by the *erm41* typing method and it was treated as correct when either of them was identified from the comparing methods.
[c]NA, not applicable.

## DISCUSSION

This novel SSS method accurately and quickly differentiated the three MAG species, as well as *M. chelonae* and *M. immunogenum*, the five clinically relevant species in the MCAC. The accuracy of the SSS method was demonstrated by comparing to the Sanger sequencing method and to the *in-silico* analysis results and showed 100% agreement with *rpoB* typing method for both identifications and sequence homologies. To the best of our knowledge, this is the first pyrosequencing method with novel primers used to differentiate these species. This method identifies the species quickly (within 8 h) and costs a fraction of other sequencing methods (26). By integrating this test concurrently with other pyrosequencing tests, such as other AFB or routine bacterial identifications, the testing time and reagent costs are minimal. This testing method uses affordable instruments and an in-house database that can be easily established. The pyrosequencing technology is going to use the updated PyroMark Q24 or Q48 instruments (Qiagen) to replace the obsolete instrument PyroMark ID we described in this study. The new instrument that has the same functions is more affordable, easier to operate, and capable of auto-preparation of samples for sequencing.

This study was focused on identifying the three MAG species and was not extended to the other MCAC species. In addition to the five species that we targeted, six other species have been reported in the MCAC. These six species or subspecies are mostly recovered from water-related sources and are less likely to be clinically relevant (3). However, they were evaluated *in-silico* from their type strains with the same primers that we used in this study (Fig. 2). The *in-silico* analyses indicated that the other six species would produce PCR amplification products. While five of these species also are expected to have sequencing products, *M. franklinii* is anticipated to lack a sequencing product due to a 9-bp long insertion at the binding site for the sequencing primer. *M. saopaulense* would have its own distinctive signature sequence that allows it to be

| Organism | *In-Silico* Signature Sequence | References |
|---|---|---|
| *M. abscessus* | **GGGTGAGC**CCGAGTGGGCG**GCC**AA**T**CT**G**CC**T**G**CA**G | (12, 33) |
| *M. chelonae* | `---CA-C-----------CAG--C--C--C-AG-` | (34-35) |
| *M. salmoniphilum* | `---CA-C-----------CAG--C--C--C-AG-` | (36) |
| *M. stephanolepidis* | `---CA-C-----------CAG--C--C--C-AG-` | (37) |
| *M. chelonae* subsp. *bovis* | `---CA-C-----------CAG--C--C--C-AG-` | (24) |
| *M. chelonae* subsp. *gwanakae* | `---CA-C-----------CAG--C--C--C-AG-` | (38) |
| *M. saopaulense* | `A--ATCTC----------CAG--C--C--C-AG-` | (23) |
| *M. franklinii* | (No expected pyrogram sequence produced)<br>(CGGCAACGCTGAGATTCCGGCGTGGGCGCAGAACCTCCCCGAGG) | (14) |

FIG 2 *In-silico* analysis of typical short pyrogram signature sequences (35 bp) for other organisms in *Mycobacterium chelonae-abscessus* complex (MCAC) (33–38). –, indicates the same nucleotide as that in *M. abscessus*.

differentiated from the other species. The other four species would have short sequences consistent with *M. chelonae* (collectively, *M. chelonae* group), including *M. chelonae* subsp. *bovis*, a cattle-origin organism (24). Importantly, the three MAG species can be distinctively separated from all other MCAC species according to the *in-silico* analyses.

Recent data from whole-genome or multiloci sequencing support the three distinct MAG species (*M. abscessus*, *M. massiliense* and *M. bolletii*) (6, 30), which we adopted here, though some place them as subspecies of *M. abscessus* (31). In our study, the three MAG species were the predominant clinical isolates (71%) among the MCAC. Most of the MAG isolates were *M. abscessus* (62%), which was a similar species distribution as reported recently (63%) (22). We did not detect any of the other nontargeted six species in the MCAC from our clinical laboratory.

MAG species are not typically fully identified in most clinical laboratories, and thus may cause difficulty for clinicians in selecting optimal antibiotics (32). Many clinical laboratories have limited access to molecular species identification methods available owing to their complexity and cost. MALDI-TOF method is becoming popular for bacterial identifications but the performance for MCAC species is inconsistent (Table 3) and needs further development (15–17). MALDI-TOF may require cumbersome tube-extraction procedures and success rates vary for AFB identifications (Azad, K., personal communication). This SSS method provides a useful option in differentiating the MAG species quickly with better access due to its simplicity and low cost. The technology could be an option for many clinical laboratories, even for those with limited resources. This method should help to better understand these species in the clinical and public health domains and improve the treatment of these infections.

We noted that this method is only useful for isolates that belong to the MCAC. Direct identification from an unknown bacterial isolate or other rapidly growing mycobacteria (RGM) has not been established with this method and may not be applicable to all *Mycobacterium* species. This method should not be used directly to identify other RGM species, although we tested 12 other RGM isolates representing six uncommon *Mycobacterium* species, and all were correctly identified (data not shown). We also did not see misidentification by this method during this study and in the multiyear postimplementation in the laboratory. Another limitation may be that this SSS method was solely based on the *rpoB* gene. According to the *in-silico* analysis, this SSS method had 100% agreement with the *rpoB* typing method. Compared with ANI method, the *rpoB* typing method had identification discrepancies from 0% to 1.5% and the *erm41* typing method had 0% to 3.0% discrepancies, depending on the species (Table 4B). These

discrepancies, although very low, should be further evaluated as to which is the correct method by using additional approaches, such as whole-genome sequencing.

In conclusion, this short signature *rpoB* gene pyrosequencing method was able to differentiate *M. chelonae* and the three clinically relevant MAG species (*M. abscessus*, *M. massiliense*, and *M. bolletii*) accurately and quickly.

## MATERIALS AND METHODS

**Mycobacterial isolates.** Six reference strains were obtained from ATCC (Manassas, VA) or were strains used by laboratory proficiency agencies (CAP, Northfield, IL, or New York State Department of Health, Wadsworth Center, Albany, NY) (Table 1). All other isolates (one isolate per patient) were clinical isolates recovered from AFB cultures of clinical respiratory specimens that were submitted to this laboratory from eastern and south-eastern U.S., but their clinical significance profiles were not tracked due to laboratory limitations. Before this study, the used clinical isolates had been identified as *M. chelonae-abscessus* complex (MCAC) using our existing 16S rRNA gene pyrosequencing method (25, 26). The isolates were subcultured to Lowenstein-Jensen slants (LJ, BD, Baltimore, MD) and incubated at 35 to 37°C with 5% to 7% $CO_2$ for 1 to 2 weeks, and then maintained on LJ slants at room temperature for up to 8 weeks.

**Design for signature sequence site and primers for PCR and pyrosequencing.** The *rpoB* gene sequence from *M. abscessus* type strain CIP 104536 (=ATCC 19977) was used as the production template for the method development in determining the sites for short signature sequencing and the primers. After the initial evaluation of the above reference sequence template, we collected and compiled multiple reference sequence files from type strains and other well-characterized strains of the five targeted species, then carried out *in silico* analysis for the sequence data using Geneious software (Geneious Prime, Auckland, New Zealand). The target sequence site was sought *in-silico* to be short (typically 35 to 50 bp), hypervariable, and adjacent to conserved regions at both ends for primer binding. After multiple *in-silico* analysis and actual preliminary experiments on some of the reference strains or isolates, a hypervariable site within the *rpoB* gene was selected and amplified using a PCR method with a pair of primers (forward: 5′-CTGAACACCCACGGTGTG and reverse: 5′-biotin-TGCTGGGTGATCGAGTAC). From the PCR-amplified fragments, the short signature sequences (typically 35 to 50 bp long) were produced using a sequencing primer on a PyroMark ID system (Qiagen, CA). The signature sequences were blasted into an in-house database for species-level identification. The database was created by compiling the refence *rpoB* gene sequences from the original MAG type strains (1) or from the curated reference whole-genome sequences of other type strains or ATCC strains in the MCAC (GenBank, NCBI-NIH, Bethesda, MD).

**PCR DNA amplification, pyrosequencing, and bacterial identification.** The procedures for genomic DNA extraction, PCR DNA amplification, pyrosequencing, and bacterial identification were previously described (25, 26). Briefly, DNA was extracted using a heated cell lysis method without further purification. The PCR amplification was performed using a Mastercycler (Eppendorf, Germany), the sequencing was performed using a PyroMark ID system (Qiagen, Hilden, Germany), and the MAG species were identified by blasting the signature sequences into the in-house sequence database as described above using Identifire software (Qiagen).

**Validation study by Sanger sequencing.** To validate the SSS method identifications to the species levels, a 40-isolate subset covering the five targeted species was selected from the previously tested isolate pool and were subjected to the Sanger sequencing method identification (Table 3). The Sanger sequencing was carried out in this study by using the same protocol and primers that were used to establish the novel species taxa *M. massiliense* and *M. bolletii* (1, 5). The *rpoB* gene fragment was amplified using the primer pair of Myco-F (5-GGC AAG GTC ACC CCG AAG GG) and Myco-R (5-AGC GGC TGC TGG GTG ATC ATC). The PCR products were verified on an E-gel (Thermo Fisher Scientific, Waltham, MA) and purified using a QIAprep Spin Miniprep Kit (Qiagen, Germany). Sanger sequencing was performed in the Genomic Core Facility, University of Maryland (Baltimore, MD) on an ABI Prism 3730XL 96-capillary DNA analyzer (Applied Biosystems, Foster City, CA), using the manufacturer-recommended settings for both forward and reverse primers. The Sanger sequences were aligned to the reference sequences from type strains for the in-house database as described above using Identifire software (Qiagen) for sequence homology analysis. The final sequence alignment pairing results were manually evaluated for individual sequence quality and differential power. The Sanger sequencing species identifications were compared to the SSS method.

**Other confirmatory identifications.** The following methods were evaluated for the identifications: (i) salt tolerance test: a subset of 50 isolates from the tested isolate pool were incubated on 5% NaCl LJ slants (Remel, Thermo Fisher Scientific, Waltham, MA) for 2 weeks to determine the salt tolerance (13); (ii) MALDI-TOF mass spectrometry was performed on a subset of 42 isolates using the tube extraction method according to the manufacturer's instructions (Bruker, Bremen, Germany).

**In-silico analysis for the species identifications.** In addition of the SSS method validation by Sanger sequencing method as described above, the species identification accuracy by the SSS method was further examined by using *in-silico* analysis. This was done on the diverse but well-characterized strains of the MAG species that were grouped by Minias et al. (22). This SSS method was applied to the strains that were selected randomly from the organism pool and the results were compared with reported data from methods of average nucleotide identity (ANI), *rpoB* typing, or *erm41* typing, while the whole genomic-based ANI method was used as the reference method. The curated reference sequences of selected strains were retrieved from GenBank (NIH) and their *in-silico* PCR and pyrosequencing products were generated based

on this SSS method as described above using software Geneious Prime (Biomatters, New Zealand) or BioEdit (University of North TX, USA). The *in-silico* sequence products were examined and then blasted into our in-house database as described above to obtain the identification results that were compared with the reported ANI or *rpoB* typing results.

Two phases of the SSS method *in-silico* analysis were conducted to compare both the identifications and their sequence homologies. First, the short signature sequence products of this study were compared with the *rpoB* gene sequences from their respective type strains. Second, the SSS method *in-silico* results were compared with the ANI or *rpoB* typing results of randomly selected or manually picked (for those having result discrepancies with either or both *erm41* typing and ANI method) strains among the 1505 MAG species strain pool as mentioned above (22).

## SUPPLEMENTAL MATERIAL

Supplemental material is available online only.

**SUPPLEMENTAL FILE 1**, XLSX file, 0.02 MB.

## ACKNOWLEDGMENTS

We thank the technologists on the pyrosequencing bench, Lynn Eklund for technical assistance, and Andrew Hellman of Quest Diagnostics for his critical comments on the manuscript.

Authors are Quest Diagnostics employees.

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
