## [Reviewer comments · Microbiology Spectrum]

Microbiology Spectrum

Short Signature *rpoB* Gene Sequence to Differentiate Species in *Mycobacterium abscessus* Group (MAG)

Jian Rong Bao, Kileen Shier, Ronald Master, Robert Jones, and Richard Clark

Corresponding Author(s): Jian Rong Bao, Quest Diagnostics

Review Timeline:

Submission Date:	December 21, 2021
Editorial Decision:	March 28, 2022
Revision Received:	May 17, 2022
Editorial Decision:	July 5, 2022
Revision Received:	July 8, 2022
Accepted:	July 18, 2022

Editor: Digby Warner

Reviewer(s): The reviewers have opted to remain anonymous.

Transaction Report:

DOI: <https://doi.org/10.1128/spectrum.02534-21>

March 28, 2022

Dr. Jian Rong Bao
Quest Diagnostics
14225 newbrook dr
Chantilly, Virginia 20151

Re: Spectrum02534-21 (Short Signature rpoB Gene Sequence to Differentiate Species in Mycobacterium abscessus Group (MAG))

Dear Dr. Jian Rong Bao:

Thank you for submitting your manuscript to Microbiology Spectrum. Your paper has been considered by two expert reviewers who agree that the paper describes a potentially useful approach to differentiating species within the Mycobacterium abscessus complex. However, both reviewers have identified specific revisions which are required to ensure the results are interpreted adequately and that the potential utility of this approach (and its limitations) are fairly presented. In particular, concerns were raised that the potential limitations of the approach are inadequately acknowledged, and that the number of species tested (especially *M. massiliense* and *M. bolletii* genomes) is small and should be expanded.

Link Not Available

Sincerely,

Digby Warner

Journals Department
Reviewer comments:

Reviewer #1 (Comments for the Author):

This study was aimed at developing a rapid pyrosequencing assay focusing on one target gene (*rpoB*) to discriminate between the three subspecies of the Mycobacterium abscessus complex (MABC) in addition to the two main species of the *M. abscessus*-*M. chelonae* complex. The authors argue that this assay will improve identification and treatment of infections caused by the members of this complex. Several studies have already been published describing approaches to discriminate between

the different species of *M. abscessus* including *rpoB* sequencing, therefore this work is not entirely novel. However, being able to reliably identify the three subspecies of MABC is important mostly for predicting the susceptibility to macrolides that are key drugs for the treatment of infections caused by MABC. Two of the MABC species (*abscessus* and *bolletii*) are generally resistant to macrolides whereas the third species (*massiliense*) is susceptible. In this regard, the work presented in this manuscript is relevant.

The manuscript itself is at time lacking organization and clarity. Details are missing, particularly regarding the in-silico methodology leading to the choice of *rpoB* as a target and the identification of the target signatures and should be provided in Methods. There are some significant issues with this work. It is unclear how the assay was validated after being developed and before being utilized to test the collection of 111 clinical MCAC isolates. There is a lack of a gold standard identification method to which the results obtained with could be compared to determine the sensitivity and specificity of the pyrosequencing assay. It seems that most if not all of this part of the validation was performed in-silico with an in-house database containing publicly available *rpoB* sequences extracted from GenBank. No information is provided on what type of QC the authors performed on their database. One of the problems with using Genbank for clinical purpose is that sequences are not curated and might not always be accurate. One perfect example is ATCC strain 700868 that is labeled as *Mycobacterium abscessus* but was identified as *Mycobacterium bolletii* with the assay described in this manuscript. Were additional sequence data available for all the extracted *rpoB* sequences? This absence of a solid reference method is also a potential problem for the retrospective testing on 111 MCAC isolates. All the alternative methods of identification used in this study have some limitations and do not allow full evaluation of the accuracy of the SSS assay. Methods such as WGS, if available or line probe assay would have been preferable.

Several studies have clearly suggested that *rpoB* sequencing should not be used as a stand alone for identification of *M. abscessus* complex species mainly due to the occurrence of horizontal gene transfer between these different subspecies. This has been observed particularly with *M. massiliense* that can transfer *M. abscessus* *rpoB* and therefore could be misidentified as *M. abscessus* by the SSS assay while still being susceptible to macrolides. Recommendations are to use a two-tier approach with *rpoB* sequencing as the primary assay, followed with a secondary assay looking a different target such as *erm* (41) for example. This potential issue should have been addressed by the authors.

Other issues:

Line 38: *M. abscessus* complex species are not phenotypically indistinguishable

Line 73: *rpoB* is not a multi copy gene as stated

Line 95: It seems that the hypervariable site that is targeted is within the *rpoB* gene coding sequence and not in the promoter region as stated. Please clarify

Line 113: Should this read Table 3 instead of Table 2? In addition, in Table 3, should it read *rpoB* instead of 16SrRNA?

More generally, this table is very confusing and of limited interest and therefore should be deleted

Reviewer #2 (Comments for the Author):

The article presents a new method of identification of species of *Mycobacterium abscessus* complex. The authors indicated a 35 bp sequence and validated its usefulness for differentiation of 111 strains of *M. abscessus*-*chelonae* complex with pyrosequencing. While the method could be valuable for a broader public, the manuscript requires substantial corrections.

1. Please mention Hain Lifescience test GenoType NTM-DR for differentiation of *M. abscessus* complex. Please remove a comment that signature sequence hybridization was tested with a limited number of samples, as the method was tested in silico with over 1500 strains.

2. There are contradictory statements in the manuscript. Authors state that sequencing of the 16S rRNA lacks differential base-pairs for the MCAC species, but they use 16S sequencing as the reference method, as indicated in Table 3. It is not clear which method the authors used as their reference. Authors cited 16S rRNA in Table 3, 16S-ITS in reference 5 (doi:

10.1128/JCM.42.12.5493-5501.2004), and part of the *rpoB* in reference 1 and primer sequence (<https://doi.org/10.1099/ijs.0.63969-0>).

3. Please expand information describing the collection of strains- when and where were the samples isolated? Were they typed at the place of isolation?

4. The manuscript would be more transparent if the authors included how they found the discriminative sequence.

5. Geneious is headquartered in New Zealand. Please mention the particular tools used in this study.

6. Please include information that pyrosequencing technology is discontinued.

7. It would be of value if the authors could expand the test of their method against a larger database of strains *M. abscessus* complex. It could be done in silico, for example against the collection of strains published in <https://doi.org/10.1038/s41598-020-73607-x>. Currently, their method was tested for six strains of *M. massiliense* and four strains of *M. bolletii*.

8. English needs to be corrected, as there are a few poorly structured sentences (line 39-41, line 42-44 etc.). Line 52 "not" instead of "nor". Line 73- Did you perhaps mean that there are several alleles circulating in the population?

Staff Comments:

Preparing Revision Guidelines

Please return the manuscript within 60 days; if you cannot complete the modification within this time period, please contact me. If you do not wish to modify the manuscript and prefer to submit it to another journal, please notify me of your decision immediately so that the manuscript may be formally withdrawn from consideration by Microbiology Spectrum.

Responses (in Orange) to the reviewer's comments

Ref: Spectrum02534-21

Re: Spectrum02534-21 (Short Signature rpoB Gene Sequence to Differentiate Species in Mycobacterium abscessus Group (MAG))

Thank you for submitting your manuscript to Microbiology Spectrum. Your paper has been considered by two expert reviewers who agree that the paper describes a potentially useful approach to differentiating species within the Mycobacterium abscessus complex. However, both reviewers have identified specific revisions which are required to ensure the results are interpreted adequately and that the potential utility of this approach (and its limitations) are fairly presented. In particular, concerns were raised that the potential limitations of the approach are inadequately acknowledged, and that the number of species tested (especially *M. massiliense* and *M. bolletii* genomes) is small and should be expanded.

The concerns have been addressed in the revised manuscript [Results (page 11) and Discussion (page 14) sections]. More than 90 (including the 5 type strains) reference sequences of the three MAG species were additionally analyzed *in-silico* and agreed 100% to the *rpoB* typing method for both ID and sequence homology. The analytical results were added to the manuscript (p14), including Table 4.

<https://spectrum.msubmit.net/cgi-bin/main.plex?el=A7QF2BupW1A7DTck5F2A9ftdoz0ffk2hTUcbhEJqSYcdMAZ>

Sincerely,

Digby Warner

Journals Department
Reviewer comments:

Reviewer #1 (Comments for the Author):

This study was aimed at developing a rapid pyrosequencing assay focusing on one target gene (*rpoB*) to discriminate between the three subspecies of the *Mycobacterium abscessus* complex (MABC) in addition to the two main species of the *M. abscessus*-*M. chelonae* complex. The authors argue that this assay will improve identification and treatment of infections caused by the members of this complex. Several studies have already been published describing approaches to discriminate between the different species of *M. abscessus* including *rpoB* sequencing, therefore this work is not entirely novel. However, being able to reliably identify the three subspecies of MABC is important mostly for predicting the susceptibility to macrolides that are key drugs for the treatment of infections caused by MABC. Two of the MABC species (*abscessus* and *bollettii*) are generally resistant to macrolides whereas the third species (*massiliense*) is susceptible. In this regard, the work presented in this manuscript is relevant.

Thanks. The word “novel” was used to describe the method, including the primers, database and procedures, not necessarily a novel methodology or technology.

The manuscript itself is at time lacking organization and clarity. Details are missing, particularly regarding the *in-silico* methodology leading to the choice of *rpoB* as a target and the identification of the target signatures and should be provided in Methods.

More descriptions had been added (p6). Additional *in-silico* analysis was carried out (p8) and results were added to the manuscript (p14).

The *in-silico* methodology has been described more detailly in the paper (p6), which we think was adequate for this application study. The development that led to the choice of the targeted fragment and the primers was a gradual process when both the preliminary experiments and *in-silico* analysis were conducted during the study.

There are some significant issues with this work. It is unclear how the assay was validated after being developed and before being utilized to test the collection of 111 clinical MCAC isolates. There is a lack of a gold standard identification method to which the results obtained with could be compared to determine the sensitivity and specificity of the pyrosequencing assay.

We used three standards for the comparisons:

1. Reference strains from authentic resources, such as ATCC, or the survey agencies.

2. Use Sanger sequencing for 40-isolate subset to validate the SSS method identifications.
3. Compare the SSS method *In-silico* results from the type strains or reference strains.

To follow the reviewer's recommendations, we added extra *in-silico* analysis (88+5 type strains) from well-characterized strains, and they had both 100% agreements with the *rpoB* typing method and their sequence homologies (p8 and p11).

For the comments on sensitivity and specificity calculations, we use "agreement" for identification result category for the better fit we think (Table 4).

It seems that most if not all of this part of the validation was performed in- silico with an in-house database containing publicly available *rpoB* sequences extracted from GenBank. No information is provided on what type of QC the authors performed on their database. One of the problems with using Genbank for clinical purpose is that sequences are not curated and might not always be accurate. One perfect example is ATCC strain 700868 that is labeled as *Mycobacterium abscessus* but was identified as *Mycobacterium bolletii* with the assay described in this manuscript. Were additional sequence data available for all the extracted *rpoB* sequences? This absence of a solid reference method is also a potential problem for the retrospective testing on 111 MCAC isolates. All the alternative methods of identification used in this study have some limitations and do not allow full evaluation of the accuracy of the SSS assay. Methods such as WGS, if available or line probe assay would have been preferable.

Our in-house database was established from the type strains and reference strains, not randomly from the GenBank. The database has been validated according to the results of identification and sequence homology from reference methods (Sanger sequencing, *rpoB* typing).

The ATCC 700868 strain identification as *M. bolletii* was confirmed from the multi-gene-based Line-probe method as well as Sanger sequencing (p10).

Several studies have clearly suggested that *rpoB* sequencing should not be used as a stand alone for identification of *M. abscessus* complex species mainly due to the occurrence of horizontal gene transfer between these different subspecies. This has been observed particularly with *M. massiliense* that can transfer *M. abscessus* *rpoB* and therefore could be misidentified as *M. abscessus* by the SSS assay while still being susceptible to macrolides. Recommendations are to use a two-tier approach with *rpoB* sequencing as the primary assay, followed with a secondary assay looking a different target such as *erm* (41) for example. This potential issue should have been addressed by the authors.

A discussion was added for the potential identification discrepancies (p14).

Other issues:

Line 38: *M. abscessus* complex species are not phenotypically indistinguishable

Modified (line 56).

Line 73: *rpoB* is not a multi copy gene as stated

Removed.

Line 95: It seems that the hypervariable site that is targeted is within the *rpoB* gene coding sequence and not in the promoter region as stated. Please clarify

Modifies (p6).

Line 113: Should this read Table 3 instead of Table 2? In addition, in Table 3, should it read *rpoB* instead of 16SrRNA?

Corrected (Table 3). The last column is 16S rRNA gene for the complex ID (MCAC).

More generally, this table is very confusing and of limited interest and therefore should be deleted

The Table 3 has been re-formatted and re-organized, which should make it easier to understand. We would like to keep it as we think the information is useful.

Reviewer #2 (Comments for the Author):

The article presents a new method of identification of species of *Mycobacterium abscessus* complex. The authors indicated a 35 bp sequence and validated its usefulness for differentiation of 111 strains of *M. abscessus-chelonae* complex with pyrosequencing. While the method could be valuable for a broader public, the manuscript requires substantial corrections.

1. Please mention Hain Lifescience test GenoType NTM-DR for differentiation of *M. abscessus* complex. Please remove a comment that signature sequence hybridization was tested with a limited number of samples, as the method was tested in silico with over 1500 strains.

Removed.

2. There are contradictory statements in the manuscript. Authors state that sequencing of the 16S rRNA lacks differential base-pairs for the MCAC species, but they use 16S sequencing as the reference method, as indicated in Table 3. It is not clear which method the authors used as their reference. Authors cited 16S rRNA in Table 3, 16S-ITS in reference 5 (doi: 10.1128/JCM.42.12.5493-5501.2004), and part of the *rpoB* in reference 1 and primer sequence (<https://doi.org/10.1099/ijls.0.63969-0>).

The 16S rRNA gene was not used as the reference method but was used to identify the isolates to MCAC when the isolates were collected for the study.

We used the reference #1 & #5 so that the type strains (*M. bolletii* & *M. massiliense*) were addressed.

The reference 1 is where the new taxa *M. bolletii* was described. We used the PCR primers described in the reference for our Sanger sequencing for the method comparison, not for this SSS method itself.

The reference 5 was where the new taxa *M. massiliense* was described and multi-loci sequencing method, including partial *rpoB* sequencing, was used.

3. Please expand information describing the collection of strains- when and where were the samples isolated? Were they typed at the place of isolation?

Additional information has been added to the manuscript (p5). The isolates were identified as MCAC at the collection.

4. The manuscript would be more transparent if the authors included how they found the discriminative sequence.

Additional information was added in the manuscript (p6, as discussed above).

5. Geneious is headquartered in New Zealand. Please mention the particular tools used in this study.

Geneious Prime was used. The headquarter has been corrected (Line 115).

6. Please include information that pyrosequencing technology is discontinued.

Pyrosequencing technology is not discontinued, but PyroID has been replaced with Q48 or Q24 instruments. This was listed in line 252-255.

7. It would be of value if the authors could expand the test of their method against a larger database of strains *M. abscessus* complex. It could be done in silico, for example against the collection of strains published in <https://doi.org/10.1038/s41598-020-73607-x>. Currently, their method was tested for six strains of *M. massiliense* and four strains of *M. bolletii*.

Additional *in-silico* analysis was added as suggested and all of the SSS method has 100% agreement to the *rpoB* typing results (p11).

8. English needs to be corrected, as there are a few poorly structured sentences (line 39-41, line 42-44 etc.). Line 52 "not" instead of "nor". Line 73- Did you perhaps mean that there are several alleles circulating in the population?

Revised.

Staff Comments:

Preparing Revision Guidelines

- Point-by-point responses to the issues raised by the reviewers in a file named "Response to Reviewers," NOT IN YOUR COVER LETTER.

Done.

- Upload a compare copy of the manuscript (without figures) as a "Marked-Up Manuscript" file.

Done.

- Each figure must be uploaded as a separate file, and any multipanel figures must be assembled into one file.

Done.

- Manuscript: A .DOC version of the revised manuscript

Done.

- Figures: Editable, high-resolution, individual figure files are required at revision, TIFF or EPS files are preferred

NA.

Please return the manuscript within 60 days; if you cannot complete the modification within this time period, please contact me. If you do not wish to modify the manuscript and prefer to submit it to another journal, please notify me of your decision immediately so that the manuscript may be formally withdrawn from consideration by Microbiology Spectrum.

July 5, 2022

Dr. Jian Rong Bao
Quest Diagnostics
14225 newbrook dr
Chantilly, Virginia 20151

Re: Spectrum02534-21R1 (Short Signature rpoB Gene Sequence to Differentiate Species in Mycobacterium abscessus Group (MAG))

Dear Dr. Jian Rong Bao:

Thank you for submitting your manuscript to Microbiology Spectrum. As you will see your paper is very close to acceptance. Please modify the manuscript along the lines which Reviewer 2 has recommended (see below for full comments); that is:

- (i) including a reference to the NTM Hain Life Sciences test;
- (ii) restructuring the indicated sentence regarding NTM infections; and
- (iii) citing ANI as reference method.

As these revisions are quite minor, I expect that you should be able to turn in the revised paper in less than 30 days, if not sooner.

When submitting the revised version of your paper, please provide (1) point-by-point responses to the issues I raised in your cover letter, and (2) a PDF file that indicates the changes from the original submission (by highlighting or underlining the changes) as file type "Marked Up Manuscript - For Review Only". Please use this link to submit your revised manuscript. Detailed instructions on submitting your revised paper are below.

Link Not Available

Sincerely,

Digby Warner

Reviewer comments:

Reviewer #2 (Comments for the Author):

The Authors strengthened their research by including a reference set of well-characterized strains. The authors generally responded to comments and left minor issues of the presentation unanswered. For example, I do feel that mentioning a popular NTM Hain Life Sciences test in the introduction section would add credibility to the manuscript. Restructuring the sentence in previous line 52, to clearly define the subject to infections and not (possibly) to humans, would make the manuscript more elegant. It would probably be better to refer to ANI as a reference method than to rpoB typing from studies by Minias et al. RpoB typing showed issues due to HGT in the original research. ANI, based on WGS, is a more credible method of differentiation.

Preparing Revision Guidelines

To submit your modified manuscript, log onto the eJP submission site at <https://spectrum.msubmit.net/cgi-bin/main.plex>. Go to Author Tasks and click the appropriate manuscript title to begin the revision process. The information that you entered when you first submitted the paper will be displayed. Please update the information as necessary. Here are a few examples of required

updates that authors must address:

- point-by-point responses to the issues I raised in your cover letter
- Upload a compare copy of the manuscript (without figures) as a "Marked-Up Manuscript" file.
- Each figure must be uploaded as a separate file, and any multipanel figures must be assembled into one file.
- Manuscript: A .DOC version of the revised manuscript
- Figures: Editable, high-resolution, individual figure files are required at revision, TIFF or EPS files are preferred

Please return the manuscript within 60 days; if you cannot complete the modification within this time period, please contact me. If you do not wish to modify the manuscript and prefer to submit it to another journal, please notify me of your decision immediately so that the manuscript may be formally withdrawn from consideration by Microbiology Spectrum.

Responses (in Orange) to the reviewer's comments

Re: Spectrum02534-21R1 (Short Signature rpoB Gene Sequence to Differentiate Species in Mycobacterium abscessus Group (MAG))

Thank you for submitting your manuscript to Microbiology Spectrum. As you will see your paper is very close to acceptance. Please modify the manuscript along the lines which Reviewer 2 has recommended (see below for full comments); that is:

(i) including a reference to the NTM Hain Life Sciences test;

Two references (#19 & 20) were added in the introduction (Lines 82-84).

Accordingly, all the reference #s after the #18 have been moved 2 digits up in the manuscript.

(ii) restructuring the indicated sentence regarding NTM infections; and

The sentences have been restructured (Lines 50-54).

(iii) citing ANI as reference method.

The ANI was cited as the reference method for the *in-silico* analysis (Lines 159-163).

Accordingly, the following changes were based on the reference method:

1. A few statement modifications (Lines 237-239, & 297-299).
2. A few data in Table 4 were re-arranged or re-calculated and checked (Page 26).

As these revisions are quite minor, I expect that you should be able to turn in the revised paper in less than 30 days, if not sooner.

July 18, 2022

Dr. Jian Rong Bao
Quest Diagnostics
14225 newbrook dr
Chantilly, Virginia 20151

Re: Spectrum02534-21R2 (Short Signature rpoB Gene Sequence to Differentiate Species in Mycobacterium abscessus Group (MAG))

Dear Dr. Jian Rong Bao:

Your manuscript has been accepted, and I am forwarding it to the ASM Journals Department for publication. You will be notified when your proofs are ready to be viewed.

Sincerely,

Digby Warner
Editor, Microbiology Spectrum
